# Mechanism of collagen folding propagation studied by Molecular Dynamics simulations

**Julian Hartmann, Martin Zacharias** *

Center for Functional Protein Assemblies, Technische Universität München, Garching, Germany

* zacharias@tum.de

## Abstract

Collagen forms a characteristic triple helical structure and plays a central role for stabilizing the extra-cellular matrix. After a C-terminal nucleus formation folding proceeds to form long triple-helical fibers. The molecular details of triple helix folding process is of central importance for an understanding of several human diseases associated with misfolded or unstable collagen fibrils. However, the folding propagation is too rapid to be studied by experimental high resolution techniques. We employed multiple Molecular Dynamics simulations starting from unfolded peptides with an already formed nucleus to successfully follow the folding propagation in atomic detail. The triple helix folding was found to propagate involving first two chains forming a short transient template. Secondly, three residues of the third chain fold on this template with an overall mean propagation of ~75 ns per unit. The formation of loops with multiples of the repeating unit was found as a characteristic misfolding event especially when starting from an unstable nucleus. Central Gly→Ala or Gly→Thr substitutions resulted in reduced stability and folding rates due to structural deformations interfering with folding propagation.

**Data Availability Statement:** All relevant data are within the manuscript and its Supporting information files.

**Funding:** This study was funded by SFB 1035 (German Research Foundation DFG,

## Author summary

The extracellular matrix is stabilized by collagen, a fibrillar protein structure, which represents the most abundant protein of the human body. Collagen consists of three peptide chains that form an elongated triple helix with a repeating and largely conserved sequence pattern of two proline (or hydroxyproline) residues followed by a glycine. Several human diseases are associated with mutations in collagen. The folding propagation is the most critical step in the collagen structure formation and not well understood. We have used multiple Molecular Dynamics simulations to specifically investigate the mechanism of triple helix propagation and how it is affected by mutations. The folding propagation was found to involve first two chains forming a short transient template followed by three residues of the third chain to fold on this template. Additional simulations were used to characterize misfolding events such as loop formation and the effect of glycine substitutions on collagen folding.

Sonderforschungsbereich 1035, Projektnummer 201302640, project B02) (https://www.dfg.de/) to MZ. Computational resources were provided by the Leibniz Supercomputing Center (LRZ) within grant pr27za. The funders had no role in study design, data collection and analysis, decision to publish, or preparation of the manuscript.

## Introduction

With one third of human proteins, collagen is the most abundant protein of our body. It appears in many different tissue types like skin, cartilage, bone or hair and plays an essential role for the stability of the extracellular matrix and whole body.[1] The various types of collagens allow them to serve different functions ranging from stiff structures like bones to elastic tissue like skin or cartilage. The characteristic feature of collagens is a parallel right-handed triple helical structure which was already proposed by Ramachandran and Kartha [2], Rich and Crick [3], and Cowan and coworkers [4]. It consists of three polypeptide chains, that form a left-handed poly-proline II-type helical coil. For this conformation it is important that every third residue is a glycine (Gly)[5] resulting in the characteristic Gly-X-Y repeating unit with X and Y mostly representing proline (Pro) or hydroxyproline (Hyp). The structures of various triple-helical structures have been determined by X-ray crystallography[1]. Sequence variants are known in different collagens that are associated with several human diseases like Osteogenesis Imperfecta[6], Ehlers-Danlos syndromes[7], Alport syndrome[8,9] and many others. Mutations in collagen can result in a destabilization, misfolding or delayed folding of the collagen fiber. Understanding such defects and design of treatments requires a comprehensive understanding of the structure formation processes. Kinetic measurements of folding rates for collagen-like peptides suggest that folding consists of several steps, which include a first formation of a triple-helical nucleus followed by rapid propagation of the triple-helical structure from the C-terminus to the N-terminus in a "zipper-like" mechanism [10–13]. The rate determining folding steps correspond to the formation of a sufficiently long and stable initial triple-helical segment and the cis-trans isomerization of Gly-Pro bonds that can interrupt or prolong the subsequent propagation step [10,13]. It has been possible to determine the rate of nucleus formation for model peptide chains and to conclude that the nucleus formation is associated with a purely entropic barrier (determined by the speed with which the three strands diffusive together to form a nucleus consisting of ~3.3 tripeptide units per strand). The subsequent propagation step is too fast to be measured by the fastest available kinetic mixing experiments [12], hence, propagation (with all-trans Pro) must happen in a time regime significantly below 1 ms.

*In vivo*, collagens are formed in the Endoplasmic reticulum (ER) and involve several chaperone molecules including the collagen specific Hsp47. Initiation of the collagen folding process starts at the C-terminus and involves disulfide bonds in the non-helical procollagen part, which is cut away at the final stage.[14] Since collagens can include >1000 residues any incorrect propagation can prevent folding completion. The Hsp47 chaperone can bind to already folded parts of collagen[15] and stabilizes the folded part allowing further propagation to continue towards the N-terminus.

Experimental biophysical kinetic measurements are well suited to study the initial nucleation of triple-helix formation on model systems[12,13]. However, the time resolution is insufficient to study the propagation steps that are strongly affected by known mutations and give rise to various diseases. Molecular Dynamics (MD) simulations are well suited to study structure formation processes at atomic resolution. Several MD simulation studies have already been performed on triple-helical collagen model peptides [16–21], however, typically starting from the already folded triple helix to investigate the local dynamics and the effect of substitutions [19–21]. The triple-helix folding process has also been studied but was guided by a force to gradually move the unfolded system towards the known folded structure[20]. Such added external driving forces may, however, artificially bias the structure formation processes.

In the present study, we use multiple MD simulations starting from an already formed folding nucleus to follow directly triple-helix propagation. The diffusive formation of an initial

nucleus occurs on time scales beyond current MD simulations [12]. However, for the natively folded triple helix formation of long collagen fibrils the propagation steps are of central importance.

The simulations allow us to give an atomic detail picture of the structure formation process. We find that the folding follows a sequential process with a first transient formation of two chains forming a short 3 residue folding template. The third chain then propagates by using this segment as template for folding to complete one folding propagation cycle. We also study how an unstable nucleus affects the folding process and find that it lowers dramatically the number of successful folding simulations but if it successfully starts folding follows the same mechanism as observed for the simulations starting from a stable nucleus. Finally, we also characterize misfolding events and study systematically the substitution of a central Gly by Ala or Thr on the folding propagation.

## Results and discussion

The simulation model system consists of three chains (Gly-Pro-Pro)$_5$ forming a collagen type triple helix (Figs 1 and S1). In order to specifically study the propagation during MD simulations we added an artificial harmonic restraint to the two residues at the C-terminus of each strand (only acting on the backbone C$_\alpha$ coordinates) to keep this nucleus on average reasonably close to the structure in a folded reference triple helix. It mimics an already formed triple helical nucleus structure at the C-terminus still allowing fluctuations of the restrained segment. Care was taken to adjust the flexibility to a level comparable to the fluctuations of a regularly folded triple-helical structure (S2 Fig).

Multiple folding propagation simulations of the (Gly-Pro-Pro)$_5$ structure were performed using in each case different unfolded starting structures (Table 1). Out of 10 simulations 9 successfully reached the completely folded triple helix within < 1 μs simulation time. The finally reached state is indistinguishable from a simulation started from the folded conformation (S1 and S3 Figs). Once folded the triple helix is stable for the rest of the simulation. In contrast to a mostly two-state folding of globular proteins with little accumulation of stable intermediates, the folding propagation process of the collagen triple helix follows a stepwise process (Figs 1, 2, S4 and S5). Starting from the C-terminus the root-mean-square deviation (RMSD) with respected to the folded state of the following segments decreases progressively along the strands with intermediates that differ in length by ~3 residues until the complete triple helix has formed.

Similarly, the near-native contacts increase progressively over time with occasional steep rise during folding once a propagation step is completed (Fig 2). The time between each successive folding progression of ~3 residues varies between a few ns and a few hundred ns (Figs 2 and S4 and S5).

If one looks separately at the backbone dihedral transitions of the individual chains another repeating pattern becomes visible: In a folding propagation step a segment of three residues in two strands typically forms first a transient associated native-like structure with the residues from the third chain still unfolded. The short two-strand segment forms a template or binding site for the third chain to fold along the template to complete the folding of a 3-residue segment for all three chains.

This mechanism is illustrated in Fig 3 as a sequence of successive backbone dihedral transitions. In the unfolded state a broad range of Ψ/Φ dihedral angle combinations (including near-native states) are sampled but for a folded segment only a narrow range of Ψ$_{Pro}$/Φ$_{Gly}$ combinations characteristic for a triple helix are sampled (Fig 3).

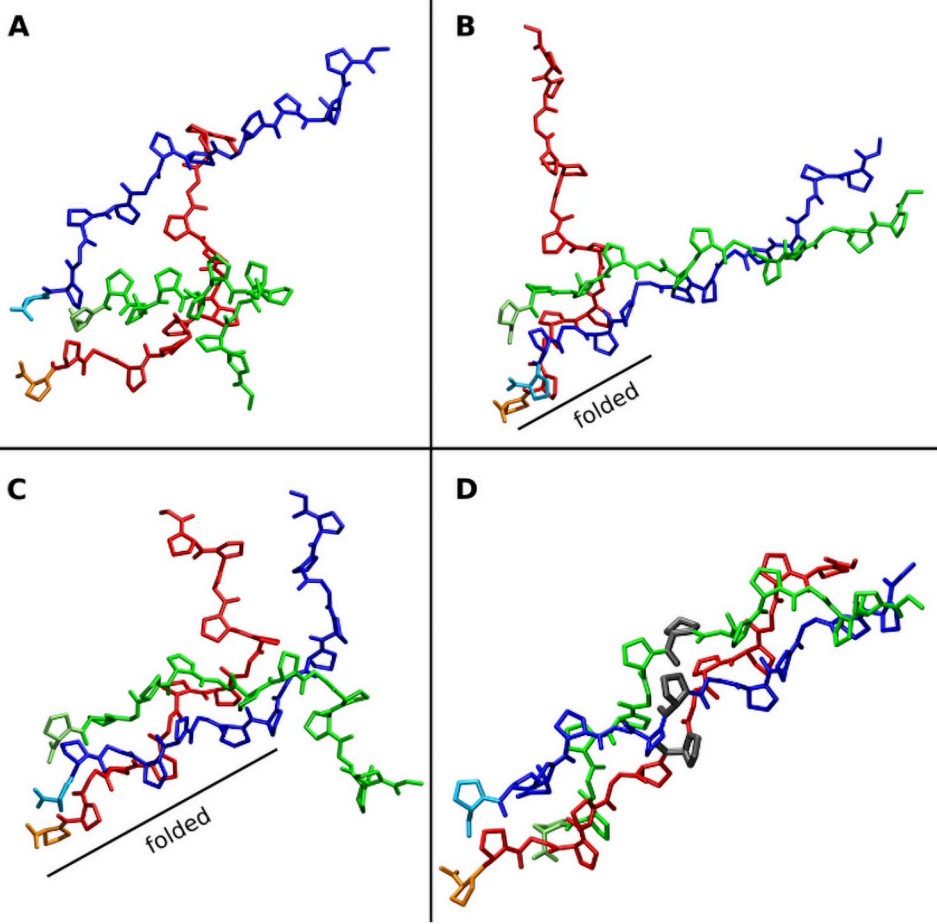

**Fig 1. Representative snapshots from a collagen folding propagation simulation.** (A) Initial random arrangement of the three (Gly-Pro-Pro)$_5$ chains (colored red, green and blue). (B) Folding nucleus formed at the C-terminus due to positional restraints on C$_\alpha$ atoms of the last two residues of each strand and partial formation of a near native structure of two chains (blue and green). (C) folding propagation with a near native arrangement of all three chains formed up to the middle of the complex. (D) Completely folded collagen triple helix (the 3 residues indicated in grey have the same residue number in each strand, are spatially close in the native structure and represent a residue triplet).

Furthermore, on another example the process is illustrated by recording the pairwise RMSD of the 7$^{th}$ residue of two selected chains (S5 Fig). Here, first chain B and C align to each other (resulting low and stable RMSD of the 7$^{th}$ residue relative to the native triple helix) but they unfold again (chain A is not folding). However, at a later time point chain A and C align to each other followed shortly later by folding of chain B on this template to the stable triplet positon including complete formation of all native contacts between the chains (S5 Fig). This example demonstrates that each chain alone or parts of it can adopt the correctly folded state several times before it eventually reaches the stable state next to the two other chains. Thus the population of dihedral angles in the "folded" state in Fig 3 also in the unfolded segments (red background) can be explained by the transient folding of individual chains. The transient formation of near native structures for individual chains is also illustrated in S6 Fig. In short time intervals of few ns individual chain can indeed reach an RMSD (non-hydrogen atoms) $< 1.5$ Å relative to the native structure.

**Table 1. Root-mean-square deviation and folding time of triple helices starting from 10 different start structures**

| simulation | wild-type[b] Å (ns) | weak restraints | G7aA Å (ns) | G7aT Å (ns) | G7abcA Å (ns) | G7abcT Å (ns) |
|---|---|---|---|---|---|---|
| 1 | 1.4 (44) | 14.7 (-) | 1.6 (879) | 5.9 (-) | 3.9 (398) | 4.8 (677) |
| 2 | 2.5 (960) | 10.1 (-) | 6.8 (-) | 3.0 (475) | 10.8 (-) | 15.4 (-) |
| 3 | 1.6 (319) | 1.6 (706) | 7.6 (-) | 5.0 (-) | 10.4 (-) | 9.2 (-) |
| 4 | 1.4 (138) | 2.5 (139) | 9.4 (-) | 2.9 (280) | 3.9 (907) | 4.8 (-) |
| 5 | 1.5 (96) | 11.2 (-) | 2.4 (769) | 2.9 (494) | 8.4 (-) | 18.7 (-) |
| 6 | 1.4 (357) | 9.0 (-) | 13.3 (-) | 1.9 (992) | 4.2 (83) | 6.2 (-) |
| 7 | 1.7 (331) | 7.9 (-) | 4.6 (-) | 9.8 (-) | 4.4 (555) | 13.9 (-) |
| 8 | 7.8 (-) | 13.3 (-) | 11.8 (-) | 1.7 (494) | 8.6 (-) | 5.1 (-) |
| 9 | 1.4 (95) | 1.9 (89) | 2.0 (905) | 3.1 (428) | 3.7 (837) | 14.1 (-) |
| 10 | 1.6 (45) | 1.7 (390) | 2.7 (635) | 6.2 (-) | 4.0 (540) | 9.1 (-) |

[a] N indicates the starting structure or simulation number.

[b] Each column corresponds to the wild type or mutated collagen sequence; G7aA means mutation only in chain A, G7abcA: mutation in every chain; RMSD is given in Å vs. native triple helix structure for the finally sampled frame (at 1 μs). In brackets the time (in ns) is given, after which the peptide was (first) completely folded.

The grouped propagation process arises from the repeating sequence of each chain (schematically illustrated in Fig 4). Since every third residue is a Gly, these are the most flexible points of the sequence. The backbone dihedral angles of the Pro residues are nearly unaltered comparing folded and unfolded peptides except for the Ψ angle at the connection to a Gly. For Gly the Φ angle differs most significantly between unfolded and folded state (S7 Fig). Taking all successful folding simulations into account the folding propagation was on average completed after ~290ns. This translates to an average formation time of one repeating 3-residue

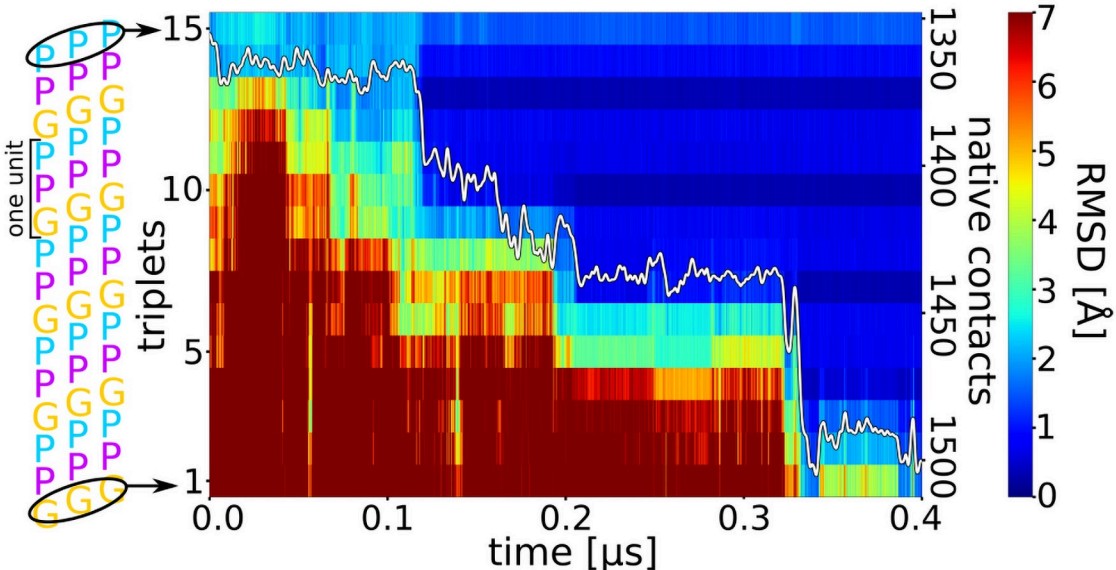

**Fig 2. Time course of triple helix formation during an MD simulation.** The simulation starts from an unfolded collagen peptide with an already formed nucleus at the C-terminus (see Fig 1A). Each labelled stripe (1–15) represents a residue triplet (residues with same number in each chain, indicated as black ellipse in the left panel) along the three strands. The root-mean-square deviation (RMSD) of each residue triplet relative to the native folded structure is indicated by a color-code (color bar on the right side). A blue color represents sampled states close to native, whereas red color corresponds to an unfolded triplet structure. The y-axis on the right side of the plot gives the number of formed native contacts (white line in the plot).

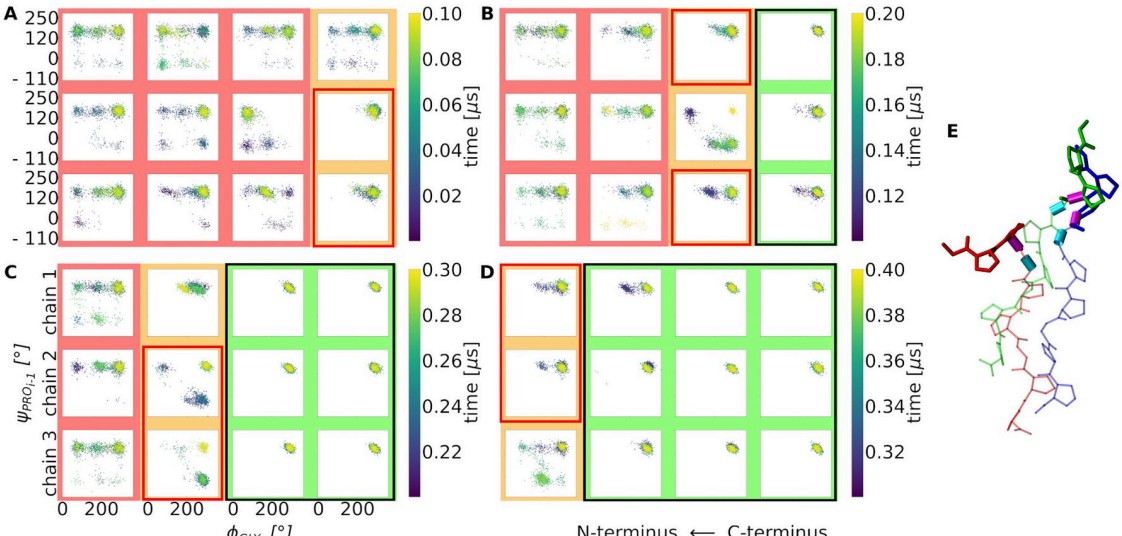

**Fig 3. Sequential collagen folding process in terms of successive dihedral angle transitions.** (A-D) 2D-Ramachandran type plot of the most flexible backbone dihedral angles ($\phi$ of $Gly_i$ and $\psi$ of neighboring $Pro_{i-1}$) in each chain of the triple helix during subsequent time periods of a folding simulation (simulation time increases from A to D). The already folded part of the triple helix, starting from the right side (C-terminus) is framed in black and highlighted by a green background (in A-D) with both dihedral angles sample the native conformation in the upper right corner of the plots. In contrast, dihedral angles cover a broader distribution in the still unfolded, disordered segments of each chain (red background). The part in between (orange) indicates an intermediate state, where two chains adopt already a native dihedral angle configuration while one is still in a non-native configuration. The process is repeated in a step-wise manner in A-D in the direction of the N-terminus of the chains. (E) Simulation snapshot indicating a partially folded triple helix with two chains overhanging by three residues in near-native geometry (blue and green) and one strand with the corresponding three residues still unfolded (red sticks). The rotatable bonds that define the $\phi$ of $Gly_i$ and $\psi$ of $Pro_{i-1}$ are shown as cyan and magenta cylinders, respectively.

unit (forming basically the elementary propagation step) of ~75 ns. It also justifies our maximum simulation time of 1 μs that is ~15 times longer than the elementary folding step.

Reducing the restraints on the backbone $C_\alpha$ atoms at the C-terminus (to 0.05 kcal·mol$^{-1}$·Å$^{-2}$, which allows in principle fluctuations of the restraint atoms by up to 4 Å within a mean energy change of RT = 0.6 kcal·mol$^{-1}$, R: gas constant, T: temperature of 310 K) led to highly mobile C-terminal start arrangements and to a significantly smaller fraction of successful folding propagation processes (Table 1 and Figs 5 and S8).

It indicates the importance of a stable folded nucleus close to the consensus triple helix near the C-terminus for further propagation. In this case only 3 of 10 starting arrangements folded

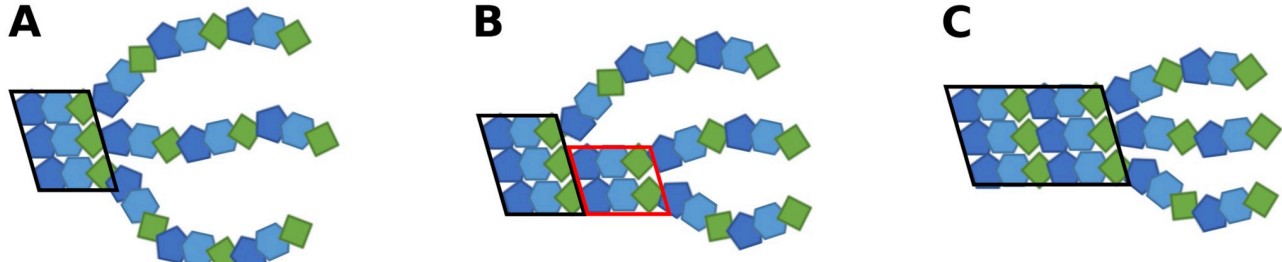

**Fig 4. Schematic illustrations of the sequential folding propagation process.** (A) Starting with an already folded part of the triple helix (framed by a black box in A, each residue is represented by a colored bead). (B) Followed by the formation of a near-native structure of two chains (red box in B) and subsequent folding of the third chain (framed box in C) to form an extension of the triple helix by three residues of each chain.

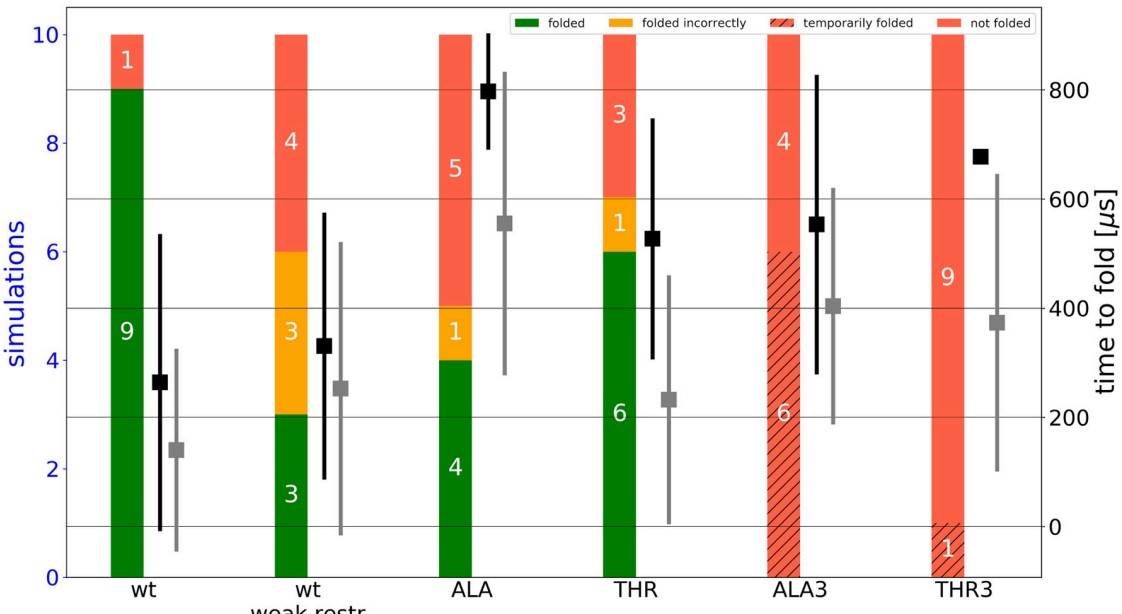

**Fig 5. Efficiency and folding times of all simulations.** Each column includes 10 simulations. Successful simulations are colored green. Simulations reaching only partly helical and stable folded peptides are colored orange and all simulations which resulted in an unfolded peptide are colored red. Red hatched bars indicate simulations which temporarily folded but subsequent unfolding. Black squares represent the average time to observe a completely folded triple helical structure. Grey squares represent the same time to result in folding up to half of the fully folded structure. Weaker restraints (positional restraints with respect to the native structure on the $C_\alpha$ atoms of the last 2 residues of 0.05 kcal·mol⁻¹·Å⁻² vs. 0.5 kcal·mol⁻¹·Å⁻² for the standard set of simulations) do not modulate the folding propagation time but the efficiency and the appearance of misfolded structures. A single point mutation of G→A/T at the center of one chain already increases the folding time and lowers the efficiency. Three point mutations of G→A/T (one at the center of each chain) do not result in a single successfully and completely folded triple helix.

correctly to a structure in close agreement with the experimental reference triple helical structure (Fig 5). However, for the successful trajectories that formed a stable nucleus with a growing tip the same stepwise mechanism and similar folding propagation times were observed (Figs 5 and S8). It emphasizes the importance of a correctly folded and stable nucleus close to the consensus triple helix near the C-terminus. Once a stable nucleus has formed stepwise propagation can proceed rapidly as has been indicated also in previous experimental studies [12]. Nevertheless, especially in the simulations with a highly mobile C-terminus several transiently stable misfolded and intermediate conformations were observed. It includes the formation of bulged loops formed by one strand during propagation steps. Interestingly, the loop regions always consist of multiples of three residue segments. Hence, three or a multiple of three residues (ending with Gly) can loop out and the triple helix eventually continues following the consensus structure after shifting the alignment with respect to the looped out strand (Fig 6A–6C). Accidental formation of such a bulge loop is critical because if propagation eventually proceeds there is no way to resolve this misfolded loop.

The fibril is on average kinked at the bulge (Fig 7). This can of course strongly affect its ability to associate with other fibrils to form a stable bundle.

In other cases, an unstable nucleus resulted in a non-native association of the chains during propagation that was stable for the rest of the simulation (Fig 6D and 6E). An unstable C-terminal nucleus also results frequently in another type of misfolded triple helices that are locally arranged in a native-like triple helical structure but the chains are shifted relative to each other.

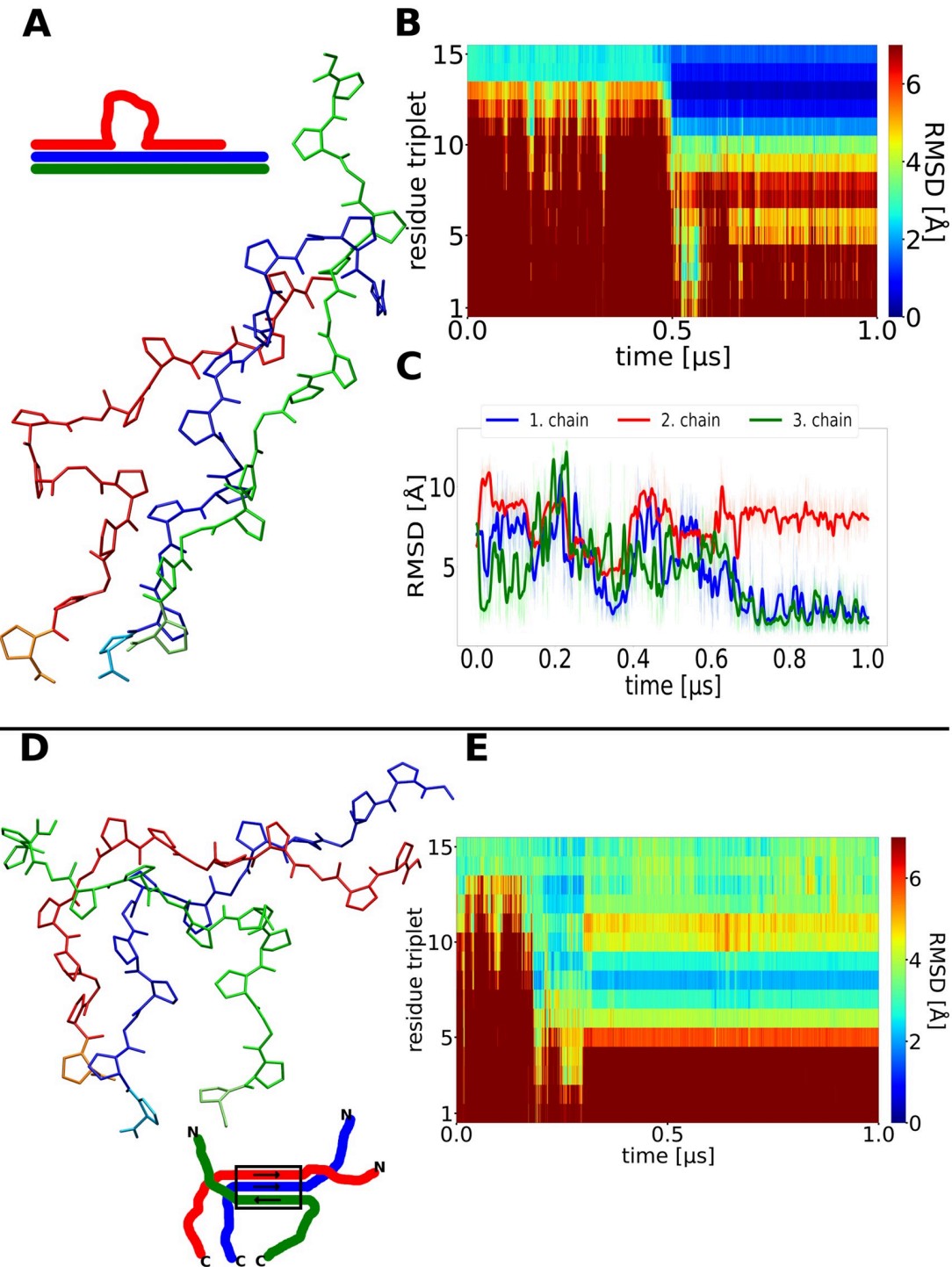

**Fig 6. Misfolding of collagen triple helices during simulations** (A) Snapshot of a finally sampled conformation with one chain (red) forming a loop but near-native conformation of the neighboring segments observed in a simulation that did not reach the native fold. (B) Time evolution of the RMSD of triplets along the simulation, after 0.5μs a correct folding of a first segment up to the loop is observed (blue regime in the plot). (C) RMSD with respect to the native folded triple helix of each chain vs. simulation time. (D) Snapshot at the final stage of a simulation with very weak restraints to stabilize the initial folding nucleus at the C-terminus of the triple helix. The green chain has detached from the other chains at the C-terminus and binds to the other two chains in a non-native arrangement. (E) The non-native arrangement remains stable for the rest of the simulation (after ~0.4 μs).

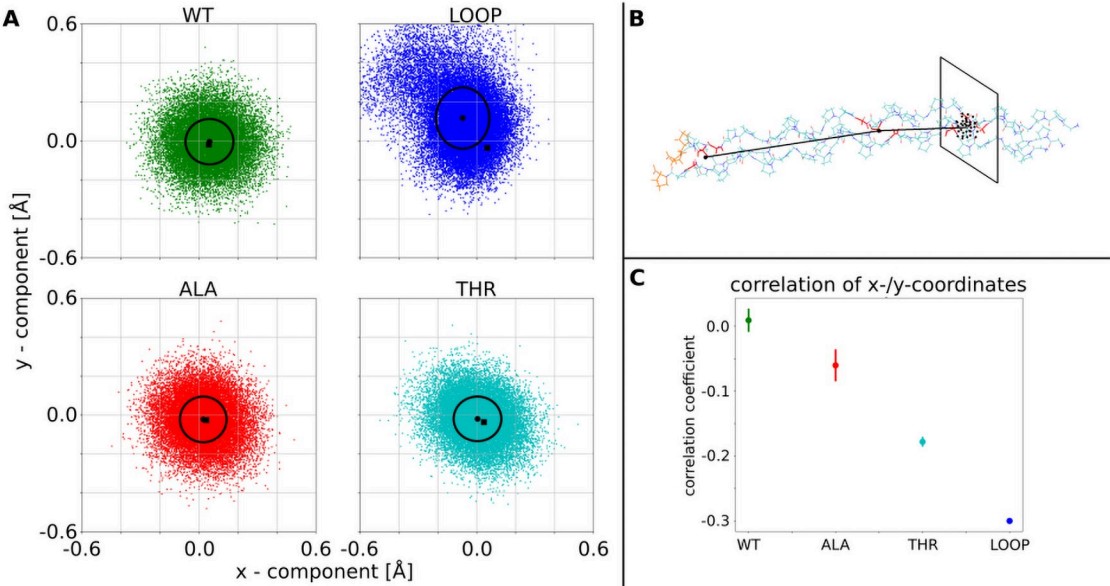

**Fig 7. Influence of central G→A/T substitution in one chain or loop in one chain on the flexibility of the triple helix.** (A) Each point in the 4 plots is the projection of a vector from the center of the folded triple helix to the N-Terminus on the plane perpendicular to the vector pointing from the C-Terminus to the center of the triple helix. WT indicates the natively folded triple helix with an isotropic distribution indicating a fluctuation around the straight triple helix (standard deviation of the distribution indicated as black circle). The LOOP, ALA, THR label indicate simulations with a central loop segment of one chain (see also Fig 6A), a central G→A or G→T mutation in one strand, respectively. The LOOP case indicates a significant directional bending fluctuation. (B) illustration of the vectors and projection. (C) correlation of x/y-displacement for the wild type, the single G7aA and G7aT mutations and the triple helix with the looped out segment in one chain. Increased correlation indicates anisotropic bending fluctuations of one end of the triple helix relative to the other end.

These shifts occur generally in units of the sequence repeat (example cases observed in simulations with an unstable nucleus are illustrated in Fig 8).

A misfolded stable structure with a loop was only observed once in the case of reasonable stabilization of the initial folding nucleus (simulation 8, see Table 1) but several times for the case of a weakly stabilized starting segment indicating again the critical importance of a stable folding start point to proceed to the correct folding propagation.

Since collagens are the product of very long genes it is likely that mutations occur at some positions and may accumulate during evolution. Mutations containing amino acids with larger side chains can lead to more significant alterations in the structure [5,10]. Since a number of mutations have been linked to important connective tissue diseases, a thorough understanding of factors that affect the folding of collagen is of particular interest [15,22,23]. Most critical is the substitution of the flexible Gly (G) by other residues and we focused or efforts on investigating the effect of G→A and G→T substitutions in either one chain or all three collagen triple helix forming chains. For instance, peptides containing a G→A mutation can form triple-helical structures that contain a local distortion at the site of the mutation and a disruption of the normal collagen hydrogen bonding pattern [24,25].

We performed 10 folding propagation simulations with one chain containing a central Gly7Ala (G7aA) or a Gly7aThr (G7aT) mutation (S8 Fig). In both cases only about half of the simulations resulted in successful propagation to the fully folded triple helix (and typically with a higher final RMSD relative to the experimental structure, Table 1). In the cases that did not propagate to the full triple helix some reached folded conformations up to the position of the substitution but the rest of the chains remained in an unfolded state (S9 and S10 Figs).

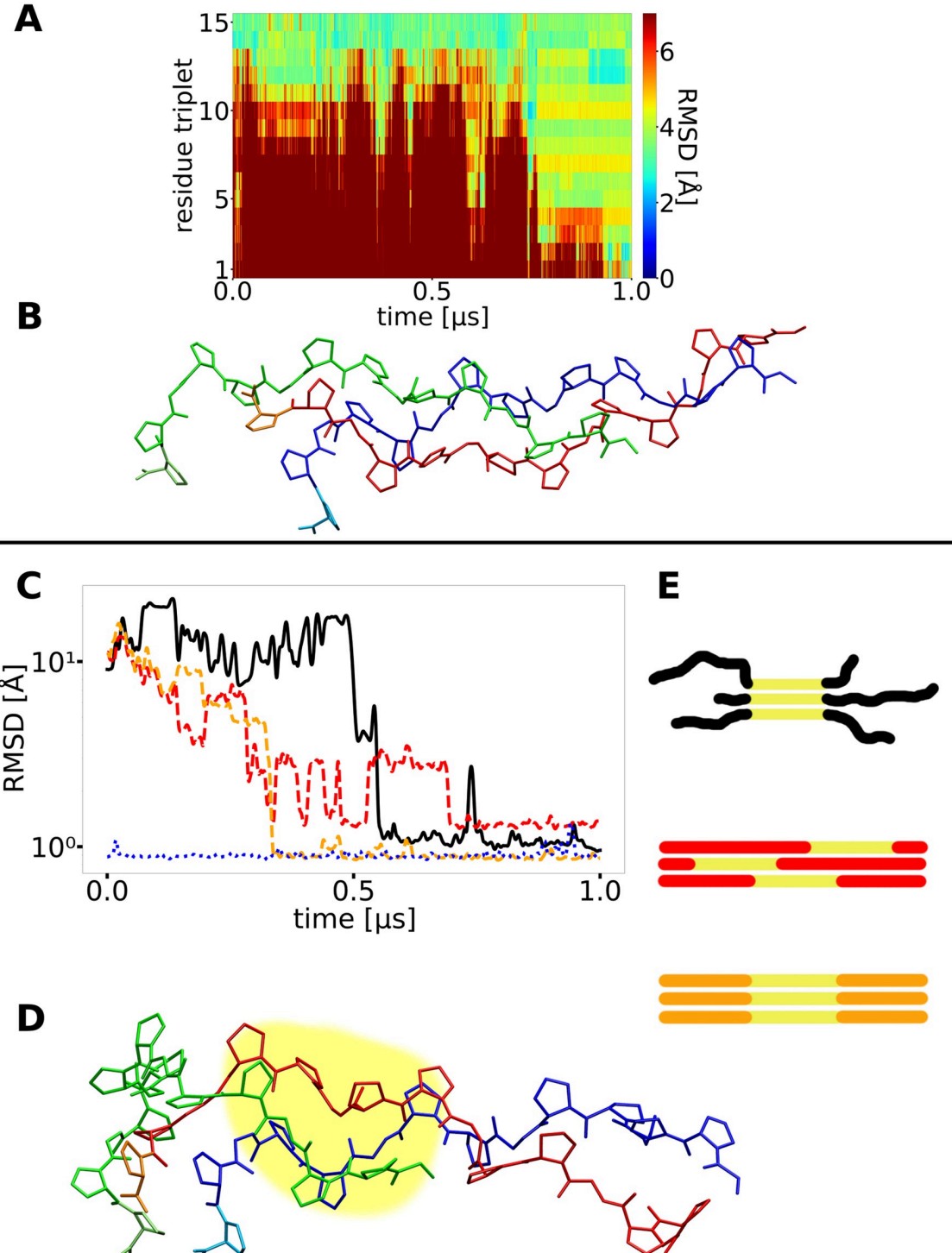

**Fig 8. Misfolding due to chain shifts.** (A) A second type of misfolding observed with very weak restraints on the C-terminal folding nucleus with the RMSD of all triplets reaching uniformly ~4 Å with respect to the native structure at ~ 1 μs simulation time. (B) A snapshot of the final conformation reveals, that large parts of the triple helix are formed, but the strands are partially shifted relative to each other causing a significant deviation of each triplet from the placement in the native structure. (C) The RMSD of segments of 3x6 residues (yellow part in I and J) of different simulations are plotted and illustrated. The black curve shows a simulation which formed a partially helix-like structure that prevented the rest of the helix from proper folding (snapshot in D and schematic illustration in E (top)). The red and orange curves show different segments of a successfully folded simulation (illustrated in E). The blue dotted example represents a helix folded from the beginning. The black curve gets close to the blue and orange one, indicating the helical structure of the yellow part. At the same time the RMSD of the same segment/residues in a folded helix is higher (red curve), the here formed structure (D) is different from the native helix.

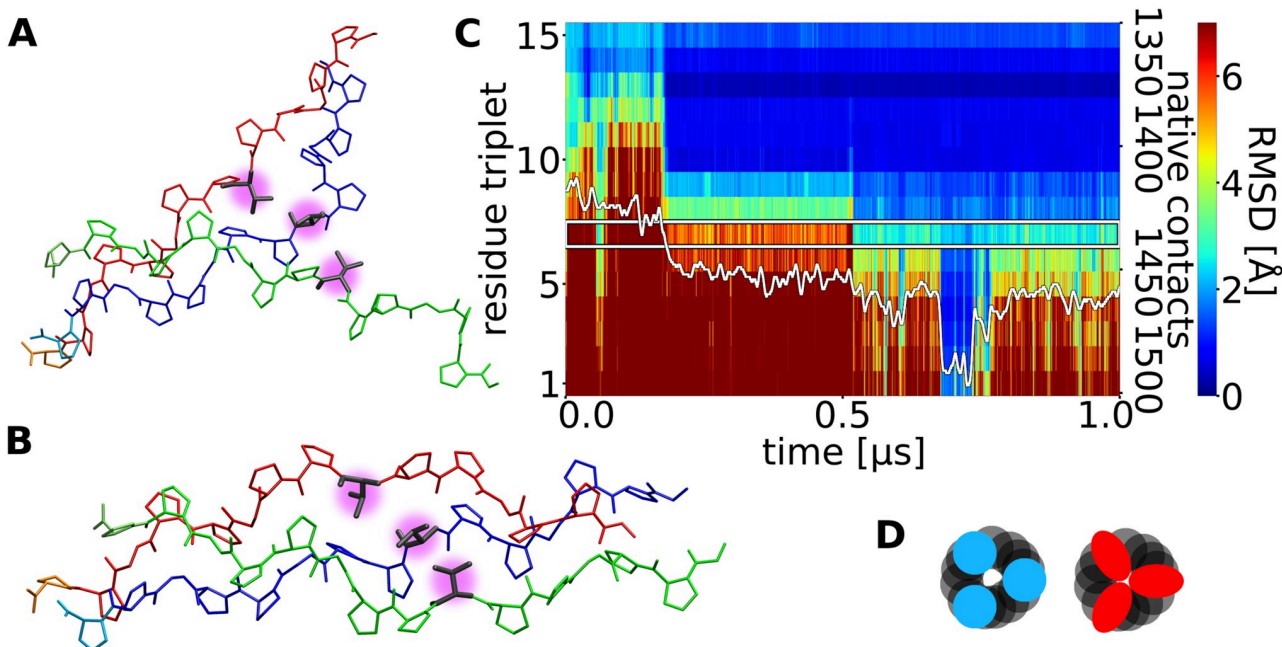

**Fig 9. Residue substitutions can strongly affect triple helix folding.** (A) Snapshot of an intermediate state with the triple helix formed up to the center of the chains that each contain a central G→T substitution (magenta sphere). (B) Short-lived sampled snapshot of the same chains, temporarily folded to a triple helical conformation, but the middle part around the mutations adopting an expanded structure. (C) RMSD plot of residue triplets along the triple helix (one residue of each chain per triplet) and native contacts. The blue part of the plot indicates correctly folded triple helix, at ~0.7 µs the complete near-native folded triple helix is observed but unfolds again at ~0.78 µs. (D) schematic cross section of helices at the position of the mutations to illustrate how the mutated residues (red) sterically collide with the other chains, compared to the WT helix (blue).

Interestingly, the average folding time in the successful cases rose to 900ns in the G7aA case and 590ns in the G7aT case qualitatively consistent with experimental result on decreased folding rates of G→A replacements in collagen model peptides [26]. Since the side chains of the mutated residues in G7aA and G7aT point inwards towards the center of the triple helix the larger side chains create a sterical barrier that hinders and delays the propagation process. Such arrangement is also seen in the X-ray structure of a collagen peptide with Gly→Ala substitutions [25].

Substitution of the central Gly in all chains at the same position finally made it impossible for the three peptide chains to fold to a stable triple helical structure (Figs 9 and S12 and S13). However, in 6 out of 10 simulations of the all G7abcA case at least transiently a structure close to a full triple helix was observed that unfolded again after a short period (~80 ns for the case illustrated in Fig 9). The result is consistent with the observed strongly reduced melting temperature of collagen model triple helices due to Gly→Ala substitutions [25,26]. For the Gly→Thr substitution in all chains only one simulation temporarily reached a folded state, none of the 10 simulated peptides stayed in this state until the end of the simulations. In this case the measured "folding times" represent the moments when the RMSD of the N-terminus was the first time at the level of the folded state. For both cases the average was in the range observed for the mutations in a single strand.

Based on an energetic analysis of the folding process using a continuum solvent model we also estimated the mean energy change during the propagation process. For the wild type case a drop of the mean energy by ~80 kcal·mol$^{-1}$ comparing the fully folded ensemble vs. the unfolded ensemble was obtained (Table 2 and S13 Fig and S1 Table). It translates to ~16

**Table 2. Mean energy difference between unfolded and folded ensembles of collagen triple helices**

| Simulation set | Wild-type[b] | Weak restraint | G7aA | G7aT | G7abcA | G7abcT |
|---|---|---|---|---|---|---|
| Mean energy change* (kcal·mol⁻¹) | -83 ± 11 | -61 ± 24 | -64 ± 19 | -72 ± 18 | -50 ± 14 | -30 ± 16 |

*mean energies were obtained using the MMPBSA (Molecular Mechanics Poisson-Boltzmann surface area method).

kcal·mol⁻¹ mean energy change upon adding one folding propagation unit to the triple helix. Note, that conformational entropy effects (that favor the unfolded state) are not considered. The substitution Gly→Ala and Gly→Thr resulted in a significant drop of the folding energy and a further strong reduction was observed for the case substitutions in all three strands (Table 2).

## Conclusions and perspective

We successfully simulated the molecular folding propagation process of collagen-like peptides during multiple simulations. Although the initiating nucleus formation of the three chains of the helix was enforced by artificial restraints our simulations can depict the process starting from a partially folded triple helix. The mechanism can be described as a zipper-like stepwise assembly of groups of multiples of three amino acids, caused by the repeating occurrence of the small and flexible Gly residue. Each repeating step-wise assembly is initiated by an approximate alignment of two chains in a conformation with the main chain dihedral angles already in the near-native regimes. This arrangement forms the template for the third chain to bind to the template for completing the propagation step of three residues in each chain. While the completed extension of all three chains combined to the triple helix forms a stable structure (a fraying of a properly folded structure was not observed) the template arrangement is unstable and can also unfold before the third chain attaches to complete the propagation step. Hence, a possible mechanism with two chains forming initially and transiently a dimeric template much longer than one repeating unit followed by folding of the third strand [13] is not supported by the simulations.

Simulations with very weak restraints on the C-terminal triple helix nucleus indicate that such unstable nucleus can result in the accumulation of misfolded structures. A looping out of segments that are multiples of the repeating unit were observed. Further propagation beyond such structures cannot be resolved and could cause formation of deformed and less stable collagen fibrils. However, even in case of weak restraints to form the nucleus for the successful folding cases a similar propagation kinetic was observed as for cases with a stabilized nucleus.

In future studies it might also be possible to follow and identify folding nuclei directly from unrestraint simulations [27]. If propagation has started the conditions on how the nucleus has been formed are not relevant. It emphasizes also the role of chaperone proteins such as Hsp47 that bind at regular intervals of an already formed collagen helix and may further stabilize the structure to allow for correct propagation.

In contrast for the simulations on chains that contain substitutions of the central Gly residue (but starting with a stable nucleus) clearly longer folding times but no increase in the type and number off misfolded conformations was found. Also in this case stabilization of already formed triple helix segments seems to be important to rapidly propagate the process. The observed longer folding times agree qualitatively with experimental results on folding of collagens with Gly→Ala substitutions [26].

## Materials and methods

The experimental structure of a collagen-like triple helix (PDB 3b0s [28]) served as starting and reference structure. This structure consists of 27 residues for each strand of the triple helix. However, in order to allow for multiple extensive folding and unfolding simulations we reduced the system to a folded triple helix consisting of three strands with 15 residues per strand with the sequence (Gly-Pro-Pro)$_5$ still including 5 repeating tripeptide units per strand. Simulations on mutations included the in silico replacement of the central Gly residue in one strand or all three strands by either Ala or Thr. For all simulations the AMBER14 [29] package in combination with the parmff14SB [30] force field for the peptides was used. To each system explicit water molecules (OPC: optimal point charge model[31]) was added in a cuboid box with 15Å distance to the molecule for peptides without any or only a single-strand mutation and 35Å distance to the molecule for peptides with mutations in all three chains. Sodium and chloride ions were added (~0.1 M) to neutralize the system. The masses repartitioning option for hydrogens and heavy atoms was used to allow for a large time step of 4 fs during simulations[32]. Long-range electrostatic interactions were included using the particle mesh Ewald method [33,34] in combination with a 9Å real space cutoff. After energy minimization (1000 steps steepest gradient method followed by 1500 steps conjugate gradient method) the systems were heated to 300 K for several nanoseconds simulation time. To generate unfolded triple helical structures the systems were heated to 700 K for 10 ns and cooled down to 310K again. To avoid trans-cis isomerization at Pro residues during this phase a dihedral restraining potential to keep a trans configuration was added. During unfolding and the folding propagation run the C$_\alpha$ atoms of the two first residues of the C-terminus were harmonically restrained to the reference structure in order to mimic a defined (but still flexible) nucleus of the triple helical folding process corresponding to an already folded triple helical segment. The force constant for this restraint was optimized at 0.5 kcal·mol$^{-1}$·Å$^{-2}$ to optimally represent the flexibility of an already formed triple helical segment (see S2 Fig). Furthermore, simulations with positional restraints at a reduced force constant of 0.05 kcal·mol$^{-1}$·Å$^{-2}$ were also performed. Each folding simulation started from a different unfolded structure and was extended to 1 µs. In addition to simulations starting from unfolded systems, simulations starting from the folded triple helix with and without residue substitutions were also performed. For each setup 10 simulations were performed. Trajectory snapshots were taken every 50 ps. Bonds involving hydrogen were constrained by SHAKE [35]. The temperature was controlled by a Langevin thermostat. To determine triple helix folding times the RMSD of the third residue triplet (meaning residue 4 of each chain) was recorded. The third residue triplet was preferred to the first one to avoid opening and closing fluctuations of the latter. A window of 50 frames (2.5 ns) was shifted through the trajectory and the first moment the mean value of the window was below a threshold of 1 Å was defined as folding time. All non-hydrogen contacts of the crystal structure with minimum distance of 4.0 Å were counted as native contacts. Energies were calculated using the MMGBSA (Molecular Mechanics Generalized Born surface area method as implemented in the Amber package. (Amber input options: PBRadii = mbondi3, igb = 8, salt concentration = 0.1M). To flatten the energy time course, a Gaussian filter (σ = 25 frames) was uses. The energy differences between the beginning and the end of a simulation are the differences of the mean energies of the first respectively last 20 frames of the simulation.

## Supporting information

**S1 Fig. Structure of a folded triple helix peptide.**
(PDF)

**S2 Fig. Root Mean Square Fluctuations of an unrestrained vs. restrained triple helix.**
(PDF)

**S3 Fig. Root Mean Square Deviation (RMSD) of individual chains vs. simulation time.**
(PDF)

**S4 Fig. RMSD of residue triplets of all wild type simulations.**
(PDF)

**S5 Fig. Detailed example of a folding process at one point along the triple helix.**
(PDF)

**S6 Fig. RMSD of individual chains during triple helix folding.**
(PDF)

**S7 Fig. Backbone dihedral angle distribution.**
(PDF)

**S8 Fig. RMSD of residue triplets of all WT simulations with reduced restraints.**
(PDF)

**S9 Fig. RMSD of residue triplets of all simulations with a single G7aA mutation.**
(PDF)

**S10 Fig. RMSD of residue triplets of all simulations with a single G7aT mutation.**
(PDF)

**S11 Fig. RMSD of residue triplets of all simulations with three mutations G7abcA.**
(PDF)

**S12 Fig. Same as S11 Fig but for simulations with three mutations G7abcT.**
(PDF)

**S13 Fig. Time course of the MMGBSA energy during three exemplary simulations.**
(PDF)

**S1 Table. Free energy differences between beginning and end of simulations.**
(PDF)

## Acknowledgments

We thank Wolfgang Wegner and Philip Marlow for helpful discussions.

## Author Contributions

**Conceptualization:** Martin Zacharias.

**Data curation:** Julian Hartmann.

**Formal analysis:** Julian Hartmann.

**Funding acquisition:** Martin Zacharias.

**Investigation:** Julian Hartmann.

**Methodology:** Julian Hartmann.

**Project administration:** Martin Zacharias.

**Resources:** Martin Zacharias.

**Supervision:** Martin Zacharias.

**Writing – original draft:** Julian Hartmann, Martin Zacharias.

**Writing – review & editing:** Martin Zacharias.

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
