## [Decision Letter · Decision Letter 0]

30 Apr 2021

Dear Dr. Zacharias,

Thank you very much for submitting your manuscript "Mechanism of collagen folding propagation studied by Molecular Dynamics simulations" for consideration at PLOS Computational Biology. As with all papers reviewed by the journal, your manuscript was reviewed by members of the editorial board and by several independent reviewers. The reviewers appreciated the attention to an important topic and the quality of the work. While minor, the reviewers have some suggestions that you may want to consider in the final version of the manuscript.

Sincerely,

Alexander MacKerell

Associate Editor

PLOS Computational Biology

Arne Elofsson

Deputy Editor

PLOS Computational Biology

[LINK]

Reviewer's Responses to Questions

**Comments to the Authors:**

Reviewer #1: The paper by Hansmann and Zacharias is a very nice MD simulation effort to follow folding propagation from a preformed nucleus in atomic detail. They apply theirs study to collagen and observe the triple helix folding propagate involving first two chains forming a short transient template. Then three residues of the third chain fold on this template. They also observe the formation of loops with multiples of the repeating unit as a characteristic misfolding event.

The paper is very nice, competently carried out and well-written. Considering the importance of nucleus formation/disruption, in the discussion the authors could just briefly comment, as a future perspective, on the possible applicability to their trajectories of methods to find folding nuclei from simulations, such as the ones described in 10.1021/acs.jctc.0c00524 or 10.1021/acs.jcim.9b00588

Reviewer #2: Understanding collagen folding and misfolding is of the utmost importance for human health, as it is associated to several diseases (especially vascular and skin diseases, but also arthritis, sclerosis and other very serious auto-immune diseases for which we do not have cures yet, but only palliative drugs that reduce pain). Its understanding is still very limited and this paper is a definite step forward.

Publish as it is with highest recommendation in PLoS Computational Biology.

This is an excellent manuscript on the collagen folding investigated with state-of-the-art molecular dynamics. It provides an accurate view of collagen fibers misfolding, in agreement with kinetic experiments.

As a minor: the authors could think changing the palette for the spectrograms, since the current palette disturbs the vision (especially in Figures 6 and 8).

Reviewer #3: Collagen is a key component of the extracellular matrix and represents the most abundant protein of the human body. Collagen exhibits a characteristic fibrillar structure, in which three peptide chains form an elongated triple helix. There is a strong sequence preference for glycine at every third position, whereas the remaining positions are mainly occupied by proline or hydroxyproline. Due to this high sequence conservation, mutations in collagen are frequently associated with diseases. However, the despite its functional importance, the details of the collagen folding pathway are only poorly understood to date.

In their manuscript, Hartmann and Zacharias have used microsecond molecular dynamics simulations to study collagen folding. The simulations were performed at high technical standard (10 copies per simulation condition) and the use of harmonic restraints for the C-terminus has been carefully evaluated.

The simulations started from a folded C-terminal nucleus, with the remaining parts of the peptide chain unfolded. By monitoring the progression of triple helix formation, the authors were able to delineate a mechanism in two chains transiently first form a template to which the third chain attaches. This process occurs sequentially (units of three residues) on a timescale of approx. 75 ns per unit. Substitution of glycine by alanine or threonine decreased the stability and folding rates of the fibril and was also a major cause of fibril misfolding.

In summary, this is a comprehensive and technically sound study, which provides a wealth of novel structural information on the collagen folding process and the mechanism by which mutations cause collagen misfolding.

**Have the authors made all data and (if applicable) computational code underlying the findings in their manuscript fully available?**

Reviewer #1: Yes

Reviewer #2: Yes

Reviewer #3: None

PLOS authors have the option to publish the peer review history of their article (what does this mean?). If published, this will include your full peer review and any attached files.

Reviewer #1: No

Reviewer #2: No

Reviewer #3: No

Figure Files:

Data Requirements:

Reproducibility:

References:

---

## [Editor Report · Decision Letter 1]

13 May 2021

Dear Dr. Zacharias,

We are pleased to inform you that your manuscript 'Mechanism of collagen folding propagation studied by Molecular Dynamics simulations' has been provisionally accepted for publication in PLOS Computational Biology.

Best regards,

Alexander MacKerell

Associate Editor

PLOS Computational Biology

Arne Elofsson

Deputy Editor

PLOS Computational Biology

---

## [Editor Report · Acceptance letter]

1 Jun 2021

PCOMPBIOL-D-21-00655R1 

Mechanism of collagen folding propagation studied by Molecular Dynamics simulations

Dear Dr Zacharias,

I am pleased to inform you that your manuscript has been formally accepted for publication in PLOS Computational Biology. Your manuscript is now with our production department and you will be notified of the publication date in due course.

With kind regards,

Kata Acsay
